# First Documentation of Life Cycle Completion of the Alien Rust Pathogen *Melampsoridium hiratsukanum* in the Eastern Alps Proves Its Successful Establishment in This Mountain Range

**DOI:** 10.3390/jof7080617

**Published:** 2021-07-29

**Authors:** Salvatore Moricca, Alessandra Benigno, Claudia Maria Oliveira Longa, Santa Olga Cacciola, Giorgio Maresi

**Affiliations:** 1Department of Agricultural, Food, Environmental and Forest Sciences and Technologies, University of Florence, Piazzale delle Cascine 28, 50144 Firenze, Italy; alessandra.benigno@unifi.it; 2Department of Sustainable Agroecosystems and Bioresources, Research and Innovation Centre, Fondazione Edmund Mach, Via E. Mach 1, 38010 San Michele all’Adige, Italy; claudia.longa@fmach.it; 3Department of Agriculture, Food and Environment, University of Catania, 95123 Catania, Italy; olgacacciola@unict.it; 4Centre for Technology Transfer, Fondazione Edmund Mach, Via E. Mach 1, 38010 San Michele all’Adige, Italy; giorgio.maresi@fmach.it

**Keywords:** alder rust, *Alnus incana*, *Larix decidua*, morphological and molecular characterization, life cycle completion, invasive pathogen, disease surveillance

## Abstract

*Melampsoridium hiratsukanum* is an alien rust fungus which has spread pervasively throughout several European countries following introduction into North Europe at the end of the 20th century. The authenticity of several records of the *Melampsoridium* species infecting alder (*Alnus* spp.) in the northern hemisphere is questionable, due to the misidentification and confusion that surround many of the older reports. Given this complicated taxonomic history, and since a *M. hiratsukanum*-like rust is strongly impacting *Alnus incana* stands in the Alps, probably affecting the bank protection role of this species along rivers, the unambiguous identification of this pathogen was a pressing epidemiological and ecological issue. In this study, field surveys, light (LM) and scanning electron microscopy (SEM), and molecular characterization were put together in an attempt to solve the conundrum. Field monitoring data, LM and SEM analyses of key taxonomic traits (length of ostiolar cells of uredinium, uredinio-spore shape and size, spore echinulation, number and position of germ pores) and ITS-rDNA sequence-based identification, convergently and unambiguously connected the rust that is causing the current epidemic to the non-native *M. hiratsukanum*. We documented the completion of the *M. hiratsukanum* life cycle on its two taxonomically unrelated broadleaf/conifer hosts. This is the first report of *M. hiratsukanum* from naturally infected *Larix decidua* in Europe.

## 1. Introduction

*Melampsoridium hiratsukanum* S. Ito ex Hirats. f. (Pucciniastraceae, Pucciniales) is a heteroecious, macrocyclic rust, which develops spermogonia and aecia on larch (*Larix* spp., conifer, aecial host) and uredinia and telia on alder (*Alnus* spp., dicot, telial host). It is not clear how this biotrophic fungus, native to Eastern Asia, reached Europe. It has been speculated that the fungus was accidentally transported with infected plant material into northern European countries during the last decade of the past century [1]. However, a possible arrival of this pathogen through natural dispersal cannot be ruled out considering the ability of rust fungi to disseminate their airborne inoculum over long-distances, and the wide geographic range of the telial and aecial hosts of *M. hiratsukanum* [2,3,4].

The chronology of reports in Europe is consistent with a natural spread of this rust from northern European countries across Central-East and Southern Europe, down to Turkey [5,6,7,8]. This sudden migration through continental Europe, across latitudinal and altitudinal gradients, was probably facilitated by the wide occurrence of broadleaf and conifer pathogen hosts that may have favored its survival and further dispersal. Pockets of *Alnus* spp., in particular, common throughout the European landscape on fens, wet woodland and along waterways, may have served as stepping-stones and natural corridors for the spread of the rust southwards [9]. *M. hiratsukanum* dispersal and establishment in non-native environments was probably also helped by intrinsic species-specific, fitness-related traits [10]. Among these are morpho-anatomical traits, such as the coriaceous, thick-walled uredinio-spores, able to resist in long and severe stressful (cold/dry) conditions, and the spine-like ornaments that uniformly cover the spore surface helping the fungus resist predation by fungivorous microarthropods [11].

In 2008, *M. hiratsukanum* was first reported in the Eastern Italian Alps [12]. Here the rust caused heavy infections, with premature leaf shedding, on the native grey alder (*Alnus incana* (L.) Moench). In this mountain range, the spring-summer period is humid and rainy, as is normal in the Alpine area, and the parasite finds these conditions optimal for its survival and reproduction. Under these conducive environmental conditions, *M. hiratsukanum* sporulates profusely on the underside of white alder leaves infected with its uredinial phase. This repeating (clonal) stage increases fungus inoculum pressure into the environment, resulting in massive rust infections which, starting from the second half of July-August, cause early defoliation and consequently weakness of the affected trees. Currently, the ecological consequences of rust infection on grey alders growing along mountain watercourses are a major cause for concern, because the disease causes loss of competitivity and is leading to the disappearance of this tree species, with a negative impact on the survival of the species-related habitat (Figure 1).

*M. hiratsukanum* is not the only rust fungus occurring on *Alnus* spp.: two congeneric rusts, *Melampsoridium alni* (Thüm.) Dietel and *Melampsoridium betulinum* (Pers.) Kleb., were also reported as infecting alder species [13,14,15]. However, the identity of the *Melampsoridium* species found on *Alnus* species in Europe in the second half of the past century is shrouded in a halo of confusion, and this was so even before the epidemic waves of the 1990s attributed to the non-native *M. hiratsukanum*. Indeed, this Asian rust had been reported decades earlier in the British Isles [16] but it was most likely a false-positive detection, as Henderson and Bennel [17] later rectified, ascribing the report from Britain to *M. betulinum*. Due to these uncertainties, the older records should be treated with caution: Hantula et al. [4,18] raised doubts about the identity of the *Melampsoridium* species reported on alders in the northern European hemisphere, arguing that some of the older records were possibly cases of species misidentification.

Given the current taxonomic confusion about the pucciniastraceous rusts infecting *Alnus* spp., and prompted by the need to accurately identify the rust taxon that is threatening the Alpine riparian formations, a study was undertaken to:(1)confirm the identity of the rust fungus, putatively identified as the alien *M. hiratsukanum*, that occurs with its uredinial and telial stages on alder trees in the eastern Italian Alps; and(2)ascertain if this *M. hiratsukanum*-like rust is the same rust that produces spermogonia and aecia on European larches (*Larix decidua* Mill.), growing in the Alps interspersed with, or adjacent to, infected alders.

The latter issue retained both ecological and mycological relevance, as larch is also a known alternate host for other heteroecious rusts that occur sympatrically in Alpine valleys. However, and more importantly, a possible confirmation that the rust that infects larch in the Alps is actually *M. hiratsukanum* would have epidemiological repercussions, because this would prove the completion of the rust life cycle in the area. Indeed, in no other part of Europe has *M. hiratsukanum* been found to infect naturally European larch [7], the development of its aecial stage on *L. decidua* having been observed only following artificial infection experiments [4].

In the present study, we used light microscopy (LM) and scanning electron microscopy (SEM) to examine the micromorphology and ultrastructure of the *Melampsoridium* rust that is plaguing grey alder formations in the area. In particular, SEM was used to analyze the fine structure of its uredinial stage, because some features of uredinia and uredinio-spores are considered important taxonomic characteristics to distinguish *M. hiratsukanum* from the two other rusts (*M. alni* and *M. betulinum*) reported on *Alnus* spp. [4,15,19,20]. Subsequently, molecular methods were used to confirm the taxonomic affiliation of this rust to *M. hiratsukanum,* as well as to prove the connection of its telial stage on *A. incana* with the aecial stage observed on *L. decidua*.

## 2. Materials and Methods

### 2.1. Field Surveys

*Alnus incana* stands have been monitored for the occurrence of rust infections since 2010, at two sites during the growing seasons. The first site was a narrow valley (Val Campelle, 46.121847 N, 11.495239 E) on the Lagorai range, in the eastern part of the Trento Province, where the disease was first reported [12]. The second site was a valley (Val di Rabbi, 46.407425 N, 10.794379E) in the Stelvio National Park, in the western part of the Trento Province. The occurrence of the fungus in these valleys was checked several times each year during the growing season, from the beginning of the growing season to late summer (second half of August), each time by accurately inspecting at least 20 alder trees per site for the occurrence of rust infection (uredinial pustules) on their lower leaf surfaces. Observations were also carried out on some *L. decidua* trees interspersed within alder stands or growing in their proximity (within a distance of approximately 50 m). *Betula pendula* Roth individuals, sometimes growing intermixed with *A. incana* in dense thickets, were also inspected for the presence of rust infection.

In addition, within the context of a survey carried out to monitor the progression of alder decline in *Alnus viridis* (Chaix.) D.C. stands [21], green alder pockets located not far from the above grey alder sites, were also surveyed for rust occurrence: in Val Maleda (46.408899 N, 10.755211 E) and in Val Sadole (46.251107 N, 11.601002 E). *Alnus glutinosa* (L.) Gaertn. Stands, mixed with grey alder or pure, were also inspected for the same purpose. 

The distribution of *M. hiratsukanum* in the Trento Province was assessed by personnel of the Forest Service of the Autonomous Province of Trento, in collaboration with FEM (Edmund Mach Foundation). The personnel of the Forest Service of the Autonomous Province of Bolzano also collected information on the presence of the fungus in alder stands in this province. 

### 2.2. Plant Sample Collection and Storage

Grey alder leaves with evident rust pustules on the lower surface were collected during surveys, when the rust was at the mature uredinial stage, with a dense cover of powdery, yellowish uredinio-spores over the lower leaf surface. Single leaves were manually collected, placed in wet polyethylene bags to maintain high relative humidity, then stored at 4 °C for a few hours, until observation in the laboratory.

Part of the infected leaf material, not to be utilized straightway for microscopic examination, was immediately stored in sterile filter paper bags. Once in the laboratory leaves were placed into a desiccator containing silica-gel, evacuated at 2–2.5 mmHg, then kept at 5 °C for 7–10 days [22], to obtain a rapid reduction of the leaf moisture content. This procedure maintains unaltered spore viability while reducing leaf overgrowing by common phylloplane fungi, among which, in particular, are some rust hyperparasites that are often associated with rust sori in the field [23,24]. 

Larch needles bearing aecial sori on their epidermis were collected with the same modalities already described for alder leaves, but in June. Infected needles were collected by holding each single needle with a tweezer, then stored in a sterile falcon tube (15 mL) kept at 4 °C during the transfer to the laboratory. 

Dried voucher specimens were stored in the herbarium of the Department of Agricultural, Food, Environmental and Forestry Science and Technology (DAGRI), Plant Pathology and Entomology Section, Florence (accession numbers IT-MH1 to IT-MH74).

### 2.3. Morphological Identification

#### 2.3.1. LM Observations 

##### Uredinial Stage on Alder Leaves

Tufts of uredinio-spores were picked off by scraping infected leaves with a dissecting needle in correspondence of the uredinia under a Wild M8 stereoscope (Leica Microsystems, Heerbrugg, Switzerland). Uredinio-spores were then transferred to glass slides and mounted in a drop of 0.5% KOH or Lactophenol for direct examination under the light microscope.

Spore morphology, surface topography and echinulation were examined under a Zeiss light microscope (ZEISS, Jena, West Germany) at ×40, ×63 and ×100 magnifications. Uredinio-spore size was determined with the same microscope by averaging 300 measurements. Images were captured with an Optikam 4083.B5 microscopic Digital USB Camera operated with OptikaView version 7 acquisition software (OptikaSrl, Ponteranica BG, Italy).

##### Aecial Stage on Larch Needles

Tufts of aeciospores were picked off by scraping infected leaves with a dissecting needle in correspondence of the aecia under a stereo microscope Nikon SM2800^®^ (Nikon Corporation, Tokyo, Japan). Aeciospores were then transferred to glass slides and mounted in a drop of 0.5% KOH or water for direct examination under the light microscope. Spore morphology and size were determined under an Eclipse 80i microscope (Nikon) dotted with digital sight images (DS-L2 imaging controller, Nikon, Melville, NY, USA).

##### Uredinio-Spore Germination Tests

Requisite amounts of uredinio-spores were taken with a fine brush from the underside of desiccated, infected leaves and rehydrated for 24 h at ambient temperature in a germinator, in a water-saturated atmosphere. Uredinio-spores were then gently brushed onto glass slides covered with a thin film of water-agar (WA) and spore viability (% germination) and the number and distribution of germ pores over the spore surface were determined at 24 h and at 48 h, again by using a Zeiss light microscope (ZEISS, Jena, West Germany) at ×40, ×63 and ×100 magnifications. Final germination percentage was determined at 48 h on 300 uredinio-spores.

#### 2.3.2. SEM Observations

To examine the ultrastructure of rust propagules under the SEM, twenty-five 1 cm^2^ leaf fragments were cut from infected alder leaves (by choosing leaf areas bearing uredinia in great profusion) and fixed in FAA (3.7% formaldehyde, 5% acetic acid, 50% ethanol). Excised leaf fragments were treated with O_4_Os for 24 h, dehydrated through sequential passages at increasing concentrations of ethanol (80%, 90%, 100%) and dried in a Bal-Tec 030 critical-point drier (CPD, BAL-TEC Balzers, Liechtenstein, Switzerland) using liquid CO_2_. Dried leaf samples were then cut into 40-μm-thick slices using a microtome cryostat (Cryo-Cut Microtome, American Optical Corp., Scientific Instrument Div., Buffalo, NY, USA). Samples were glued to special metal supports using double-sided tape and sputtered with a gold-palladium film using a sputter coater (Edwards, S-105 A, Crawley, UK). Cross-sections of uredinia, the peridial ostiolar cells, the ontogenesis of uredinio-spores within uredinia as well as the ultrastructure of uredinio-spores and their ornamentation, were examined by using a Philips 505 SEM (Philips, Gloeilampenfabrieken, Eindhoven, the Netherlands), operating at 20 kV.

#### 2.3.3. Molecular Identification

Rust-infected foliage of *L. decidua* and *A. incana* was harvested from trees growing in close proximity along riverbanks and within stands. DNA was extracted directly from alder leaves with visible uredinial pustules and from European larch needles bearing aecial sori. Uredinio-spores scattered over the leaf surface were collected with a fine tip brush using 10 mL of sterile distilled water. Individual aecia were gently excised from the needle surface and deposited in the same amount of sterile distilled water. Spore suspensions were transferred to 5 mL tubes and centrifuged for 2 min at 10.000× *g*. The supernatant was discarded and the resulting pellets were frozen at −20 °C until DNA extraction. Genomic DNA was extracted using the NucleoSpinPlant II kit (MN GmbH & Co. KG, Düren, Germany). Primer pairs ITS5-ITS4 [25] were used to amplify the 5.8 S ribosomal DNA gene and flanking Internal Transcribed Spacer (ITS) 1 and 2 regions. PCR reaction 25μL mixture included 12.5 µL of GoTaq Green Master Mix (Promega), 9.5 µL of water supplied with the GoTaq, 0.4 μM of each primer (Sigma–Aldrich) and 1 µL of DNA. Cycling was carried out in a Biometra 96 Professional thermocycler (Biometra, Germany) with an initial denaturation step at 95 °C for 5 min, followed by 30 cycles of denaturation at 95 °C for 40 s, annealing at 55 °C for 40 s, and elongation at 72 °C for 1 min, with a final elongation step at 72 °C for 10 min. Amplicons were purified using Exo-SAP (Euroclone S.p.A., Milan, Italy) and sequenced using the Big Dye terminator v3.1. cycle sequencing kit (Applied Biosystems) on an ABI Prism 3130xl Genetic Analyzer (Applied Biosystems). Nucleotide sequences were trimmed using Unipro UGENE [26], then aligned and edited using clustalW implemented in MEGA X. Sequences were processed in the NCBI database (www.ncbi.nlm.nih.gov (accessed on 1 June 2021)) with a BLAST search for matches in GenBank. Congruence of our DNA-based identification was verified by a molecular phylogeny obtained by applying Maximum Likelihood method. This analysis involved 25 nucleotide sequences. The sequences of *Melampsoridium* spp., including *M. hiratsukanum*, *M. alni* and *M. betulinum*, were chosen among the ITS sequences published in the NCBI database that presented the highest similarity with the sequences obtained in this study. Branch stabilities were evaluated by 1000 bootstrap replications performed in a NeighborJoining/BioNeighborJoining initial tree. The tree was rooted *to Cronatium ribicola*. Evolutionary analyses were conducted in MEGA X.

### 2.4. Meteorological Data 

Monthly mean temperature and monthly rainfalls for the period 2010–2020 were obtained from two meteorological stations:-Telve (Pontarso) (Meteo Trentino meteorological network), 935 m a. s. l., placed at roughly 1 Km from the study sites of Val Campelle;-Rabbi (FEM agro-meteorological network), 1444 m a. s. l., located at 1 Km from the study site of Val di Rabbi.

Only raw, non-validated, non-gap-filled meteorological data series were used. For Telve, 2010 and 2011 data were not available. 

## 3. Results

### 3.1. Field Surveys

#### 3.1.1. Rust Occurrence on *A. incana*

Rust occurrence on alder leaves was recorded yearly from 2008 in Val Campelle and from 2010 in Val di Rabbi. Infection initials appeared starting from the end of July, as single orange spots (uredinia). In subsequent weeks disease increased steadily due to leaf reinfection by uredinio-spores (Figure 2), with the leaf surface that was almost entirely covered with uredinia in late summer. 

Infection occurred throughout the crown, but its lower parts were the most severely affected; the upper parts of the crown were generally unaffected. Leaf yellowing and dropping began starting from the second week of August, reaching a peak in September. The severity of attacks was variable according to the year: in 2013, a strong rust attack was quite noticeable from the end of August, while in 2020 the occurrence of the fungus was sporadic or rare. Generally, complete yellowing or defoliation could be observed starting from the second week of September. In Val Campelle, all the surveyed trees were still alive in 2020, while in Val di Rabbi trees growing along the river showed dieback and mortality starting in 2013. On these trees, symptoms of *Phytophthora* infection were also sometimes observed. 

All the *B. pendula* trees checked for rust symptoms in the area appeared rust-free for the whole monitoring period. Rust infection was never observed on *A. viridis* trees in regularly monitored stands or in those surveyed occasionally. *A. glutinosa* trees were always uninfected, even those growing close to infected *A. incana* trees. The absence of disease on *A. viridis* and *A. glutinosa* was confirmed by the surveys of the forest services of the two Provinces (Trento and Bolzano), which reported no record of rust infections from several constantly inspected black alder and green alder stands. On the contrary, occurrence of the rust has been detected in grey alder stands of both Provinces since 2012, with the pathogen found at all surveyed sites, providing evidence of a complete colonization of the territory. 

#### 3.1.2. Rust Occurrence on *L. decidua*

In 2010, aecia were observed on the needles of larch trees growing intermixed with, or close to, infected *A. incana* trees in Val Campelle. In subsequent years, aecia were observed again on these trees in the same valley as well as in Val di Rabbi (first report in 2013), Val Vanoi (Trento province) and Val Senales (Bolzano province), in the latter first noticed in 2010, but not in subsequent years (L. Mayer, personal communication). Aecial fruiting bodies differentiated from mid-May to mid-June. Infection originated on a single needle or on a small group of needles close to each other, mainly on the lower branches of the trees. Aecial sori were sporadic, with just a few pustules developed on scattered needles, and only on those larches intermixed with, or in the immediate vicinity of, grey alders (Figure 3A,B). 

Infected needles generally dropped before the end of June. Due to the very low level of infection, needle loss by rust attacks was negligible in all years.

### 3.2. Morphological Identification

#### 3.2.1. Micromorphological Analysis

Uredinia were subepidermal, hemispherical, of average size 88.42 ± 3.79 μm, but the sizes varied from 69.81 to 103.25 μm, with a peridium which opens at the summit by a central pore made by an acute, spine-like apex, orange-yellow in color, pulverulent, mostly restricted by the veins, covering a great part of the leaf surface. Uredinio-spores appeared singly or clumped in small groups, sometimes arranged in chains, ovoid to oblong-obovoid (Figure 4A), more rarely from tear-shaped to clavate, broadly rounded at the top, measuring on average 25.8–26.3 × 13.0–13.6 μm (length × width), normally finely and regularly echinulated (Figure 4A) sometimes with echinulation (spines) sparsely and coarsely distributed over the cell wall (Figure 4B).

Uredinio-spore viability was high, with an average germination rate of 83%. Spores observed at 24 h (to monitor the emergence of germ pores) and at 48 h (to count the number and distribution of germ pores) revealed the development of multiple, bi-zonate, germ pores. Spores presented a variable number of germ pores, with some having up to six germ pores at the two apexes (Figure 4B, arrows; Figure 4C).

Aecia were sub-epidermal, cylindrical, 0.5–2 mm across, 1–2 mm erected, peridial wall colorless (Figure 3B), peridial cells from quadrilateral to hexagonal, 22–35 × 13–20 μm in size, with the peridium that ruptured at maturity to release wind-dispersed, yellowish-orange aeciospores (Figure 5A,B). Aeciospores were roundish to elliptical (Figure 5A,B), with 18.2–23.9 × 14.8–19.1 μm mean size, with a hyaline, 1.8–2.0 μm thick wall, an intense orange color and a visibly marked verrucose surface when fresh (Figure 5B).

#### 3.2.2. Ultrastructural Analysis

Scanning electron microscopy enabled an accurate view of uredinia and uredinio-spores. Under the SEM, uredinia appeared hypo-phyllous, subepidermal, hemispherical (Figure 6A), scattered or gregarious, often in groups of two (Figure 6B), with smaller peridial cells towards the apex, larger (from iso-diametrically to irregularly polygonal) in the lower portions of the sorus (Figure 6C). Ostiolar cells were ovate to conical, tapering into very long (30–45 µm), acuminate spines (Figure 6D,E), through which uredinio-spores were discharged (Figure 6E). Cross-sections of the uredinium highlighted the thick inner wall (up to 4.5 µm) of the peridium, with its internal content (uredinio-spores) (Figure 6D). Cross-sections also made it possible to examine and document uredinio-spore ontogenesis and maturation stages within the sorus, with the upper spores already mature, exhibiting a fully echinulate surface (Figure 6D,F) and the lower cells still immature (clearly detectable by their smooth, non-echinulated, surface or by echinulation just starting to form (arrow)) (Figure 6F). Urediniospores exhibited variable forms, the most common being ovoid to obovoid, with a narrower base and a broad end at the top (Figure 6G); some spores were ellipsoid, clavate or tear-shaped (Figure 6H). Uredinium and uredinio-spore sizing by scanning electron microscopy fully matched optical measurements. The high-resolution SEM imaging, contrary to the relatively low throughput optical observation, revealed that spinelike ornaments (echinulation) finely and uniformly covered the entire spore surface (also on the apex) (Figure 6G,H).

### 3.3. Molecular Identification

Both agarose gel electrophoresis and sequencing of the amplified ITS1-5.8S-ITS2 regions confirmed the amplified PCR products with gene size that ranged around 700–750 bp. Sequences searches were performed using the BLAST standard nucleotide-nucleotide basic local alignment search tool (National Center for Biotechnology Information (NCBI), Library of Medicine, Bethesda, MD, USA (http://www.ncbi.nlm.nih.gov/BLAST/, accessed on 10 May 2021)). BLAST results of ITS sequences showed a percent identity from 99.82% (accession numbers KC888944, and KC313889) to 100% (accession numbers AY394705 and KY607917) with samples of *M. hiratsukanum* sequences in GenBank. Two ITS sequences obtained in this work from uredinio-spores and aecia were deposited in GenBank with accession numbers KY576694 and KY576695, respectively. 

A phylogenetic tree based on maximum likelihood of rDNA ITS sequences of our isolates with sequences of other *M. hiratsukanum* isolates and congeneric species in GenBank confirmed our identification (Figure 7). Sequences of our isolates clustered with the other *M. hiratsukanum* isolates in the database, while they were more distantly related to *M. alni* and *M. betulinum*, clearly grouped in distinct clades. There was a total of 905 positions in the final dataset. 

### 3.4. Meteorological Data 

Meteorological data revealed high levels of precipitation during the growing season in both areas (Figure 8A,B), with a high variability in different months and years. Higher median values were recorded at the Telve meteorological station for all the months. The median value in July and August ranged between 100 and 150 mL at both sites. Fewer differences between the stations were evident in the temperature values: in July and August, they ranged between 14 and 18 °C (Figure 8B).

## 4. Discussion

All investigations, from field observations to optical micromorphology, ultrastructural analysis and molecular characterization, convergently and unequivocally identified the rust under study as the alien *M. hiratsukanum*.

Territory surveillance for rust interception revealed that the *B. pendula* trees that grew in the vicinity of rust-infected *A. incana* individuals were uninfected and this in itself demonstrates that the fungus could not be *M. betulinum* (*B. pendula* is very prone to infection by *M. betulinum* and its foliage is normally copiously covered with uredinia) [4]. Inspected *A. viridis* trees, a species widespread in the Alps, where it grows as a shrub up to 3 m, mainly at the upper limit of tree vegetation, were disease-free. *A. glutinosa* trees also were not found to bear rust fruiting bodies on their foliage in the monitored areas. Absence of rust infections on *A. glutinosa* was confirmed by similar surveys conducted in nearby Slovenia (Dusan Jurc, personal communication). Attacks on grey alders were, on the contrary, quite evident and massive: infection on the leaves, with the profuse development of the uredinial stage, was clearly noticeable from mid-July, reaching a peak in August, when the leaves started to fall.

Sequencing results showed that the isolates used in this study were *M. hiratsukanum*. Distinction of this rust from the congeneric *M. alni* and *M. betulinum* did not rely on the phylogenetic analysis, which was used only as an additional approach to highlight that the sequences belonged to *M. hiratsukanum*. In fact, widely accepted, discriminant morphological traits that can be validly utilized to distinguish *M. hiratsukanum* from *M. alni* e *M. betulinum* are: the number and position of uredinio-spore germ pores; the size, form and roundness of uredinio-spores; spore echinulation; the length of ostiolar cells of uredinia (Table 1) [4,15,19,20].

The number and the arrangement of uredinio-spore germ pores can be easily determined by germination tests [19]. Uredinio-spores of *M. alni* are known to differentiate two germ pores located at the two ends of the spore, contrary to *M. betulinum* and *M. hiratsukanum,* which are reported to have from four to six bi-zonate germ pores, positioned in pairs or groups of three on the two spore ends [20]. The new data that emerged from this study is that sometimes the number of germ pores can be less than four, having observed spores that developed only three germ tubes. However, it is also possible that some germ pores were not noticed and that some germ tubes did not develop from their respective germ pores. On the other hand, even in the study of Kaneko and Hiratsuka [19], although a number of germ tubes ranging from four to six had been reported for *M. hiratsukanum*, the image depicting germinating uredinio-spores does not show any spore with a number of germ tubes greater than three. LM examination of germinating uredinio-spores revealed that the studied rust had multiple germ pores, unlike *M. alni* that has a single germ pore at the end of each uredinio-spore. 

Optical microscope observations of all the other micromorphological characters of diagnostic value unarguably associated the studied rust with *M. hiratsukanum*. Such investigations were also substantiated by scanning electron microscopy of the ultrastructure of uredinia and uredinio-spores. *M. hiratsukanum* clearly differed from the two congeneric rusts because its uredinio-spores had different size and shape, being smaller and more rounded (ovoid to oblong-obovoid in *M. hiratsukanum*, narrower and elongated in the other *Melampsoridium* species); the uredinio-spores of *M. hiratsukanum* are completely echinulated, with ornamentations uniformly distributed over the cell surface (ornamentations occurred also on the two spore ends; these are absent in the basal and the apical parts of *M. alni* and *M. betulinum* spores); finally, *M. hiratsukanum* is clearly differentiated in having much longer ostiolar cells of the uredinium.

Similarly, micromorphological analyses of the aecial stage of the rust occurring on the needles of the conifer host (*L. decidua*) confirmed the non-indigenous *M. hiratsukanum* to be responsible for the current rust epidemic in Alpine valleys. In the identification of the aecial stage of this rust, a further element of taxonomic confusion might have been due to the sympatric co-occurrence of other rust species on *L. decidua*. Indeed, some taxa of the family Melampsoraceae also infect larch in the area and their aeciospores are morphologically similar to those of members of the genus *Melampsoridium*. This possibility was, however, ruled out because their respective receptacles are clearly morphologically different: aecia lack a peridium in *Melampsora* species whereas the peridium is well-developed (erected, cylindrical) in members of the genus *Melampsoridium* [16].

Taxonomic affiliation of this pucciniastraceous rust to the Asian *M. hiratsukanum* on the basis of micromorphological and ultrastructural analyses was further corroborated by molecular identification. Sequencing and BLAST analysis of the amplified rDNA-ITS regions from aeciospores collected on *L. decidua* definitely connected this rust to *M.*
*hiratsukanum*.

The correct taxonomic designation of this rust as *M. hiratusukanum* is further corroborated by indirect confirmation in previous culture experiments. A free-living saprophytic stage of this biotrophic pathogen on non-living media was obtained from environmental inoculum (uredinio-spore lots) collected in the same areas (Val Campelle) of this study. A sequence from an axenic culture of the fungus (GenBank acc. no. KC888944) revealed a 99% identity to two *M. hiratsukanum* isolates (GenBank acc. nos. JN581985.1 and JN581987.1) deposited in the database [27].

The identification of *M. hiratsukanum* on the aecial host *L. decidua* is important in the evolution of pathogen virulence. In fact, host-alternation (Figure 9) affords recombination (with meiosis), which promotes the development in the long-term of new variants with new combinations of virulence genes [28].

Completion of the rust life cycle on the two alternate hosts *A. incana* and *L. decidua*, demonstrates that this introduced pathogen has successfully established in the eastern sector of the Alps, where it has found a suitable habitat for its survival and reproduction. The finding of the aecial stage of the rust on a native larch species is a novelty for the European continent, natural infections of *M. hiratsukanum* having been reported in Europe only on the exotic larches *L. dahurica* Turcz and *L. sibirica* L. in the Tuusula area (Finland) [4]. Hantula et al. [4] also succeeded in inducing the formation of aecia on the native *L. decidua* following artificial inoculation with basidiospores of a fungus putatively identified as *M. hiratsukanum,* but they have never found this rust to naturally infect *L. decidua*.

*M. hiratsukanum* is rare on larch and its aecial stage is inconspicuous and of short duration. Aecia were concentrated on a few, lower branches of the trees nearest to affected alders and were not observed on larches growing far from the bottom of the valleys. Infection of larch needles was very sparse, making it difficult to observe during surveys, especially when it occurred on tall trees with a large crown. These factors might explain why the rust has not been observed on *L. decidua* elsewhere.

Host connectivity played a crucial role in Alpine valleys in the completion of the pathogen life cycle (Figure 10). 

Especially with biotrophic pathogens, a host gap constitutes an effective barrier to gamete migration and recombination. *M. hiratsukanum* is a biotrophic fungus, and the close proximity of the telial and aecial hosts favored sexual reproduction. The successful establishment and life cycle completion of *M. hiratsukanum* in the eastern Italian Alps depended on the occurrence of four favorable conditions: a marked propagule pressure (high production of rust inoculum on the telial host); host connectivity, with the broadleaf and the conifer hosts growing in contiguity; an environment conducive to disease, with high moisture during summer months, especially in July and August (meteorological data suggest the alpine environment to be highly favorable to rust survival, allowing the infection to intensify on the telial host and thus the complete colonization of this part of the grey alder range). This is different from what happens in northern European alder stands where the disease has sometimes a variable and erratic impact);larch infection is not required for rust survival: this rust can survive as mycelium in the scales of dormant alder buds [1] and thus persist in the environment regardless of the presence/absence of its aecial host (which becomes, however, crucial for the completion of the rust life cycle).

For a dozen years, *M. hiratsukanum* has been regularly observed in all grey alder stands of the monitored areas and it can be now considered fully established in the provinces of Trento and Bolzano (Trentino Alto Adige). Damage by rust infection is limited to the broadleaf host (grey alder) and results in partial defoliation, starting in the first fortnight of August. Trees generally show normal vegetation in the subsequent growing seasons but, if defoliation is severe and repeated for several years, it may result in a marked reduction of the photosynthetic activity. Impairment of photosynthesis generates negative knock-on effects: a reduced growth rate; a generalized debilitation of the trees; and, as a consequence, an increased susceptibility of alders to biotic and abiotic agents of damage [29].

The major impact of *M. hiratsukanum* infection is on riparian alder formations, as yearly repeated defoliation can weaken grey alders causing their loss of competitiveness against other species, both native and non-native. A number of invasive tree, shrub and herb species have been recorded in the riparian ecosystem of Trentino Alto Adige [30]: *Ailanthus altissima* (Mill.) Swingle, *Robinia pseudoacacia* L., *Amorpha fruticosa,* L., *Reynoutria japonica* Houtt., *Buddleja davidii* Franch., *Impatiens glandulifera* Royle, *Impatiens balfouri* Hook. f., *Solidago canadensis* L., *Heracleum mantegazzianum* Sommier & Levier, *Fallopia auberti* (L. Henry) Holub.*, Epilobium ciliatum* Raf. The arrival and establishment of this weed flora is of course a consequence of human interference, but a weakened *A. incana,* unable to maintain its dominant role in riparian ecosystem, could have facilitated the invasion process. Moreover, starting from 2011, the decline of grey alder in Trentino Alto Adige and South Tyrol was also linked to the occurrence of *Phytophthora* spp. The presence of these harmful oomycete pathogens on *A. incana* was recently confirmed in another sector of the eastern Alps, in the Veneto region [31], suggesting this new phytosanitary problem to be greater than expected. 

The pervasive spread of *M. hiratsukanum* in ecologically-impaired riparian formations has no doubt further exacerbated the decline of alder, which is particularly evident in some areas such as the Val di Rabbi. The ecological damage is significant, as *A. incana*-rich riparian formations are protected landscapes, included in the priority habitat 91E0* (Alluvial forests with *Alnus glutinosa* and *Fraxinus excelsior* (*Alno-Padion*, *Alnion incanae*, *Salicion albae*)) list of the 92/43/EEC Council Directive and play an important role further than their limited surfaces (756 ha in Trentino Alto Adige). The combined action of multiple stress factors (invasive flora and phyto-pathological constraints) could seriously compromise the resilience of these alpine fluvial ecosystems, impairing their high conservation value, functional to safeguarding soil and banks from erosion. 

## 5. Conclusions

The unauthenticity of some of the older records of *Melampsoridium* rusts on the *Betulaceae*, and on *Alnus* spp. in particular, makes them unusable for comparison. In spite of taxonomic confusion, good taxonomic characters that are sufficiently discriminant and well-established, do exist [4,15,19,20]. In this study, a combined approach, based on field observations, light and scanning electron microscopy, and molecular characterization, unarguably identified the non-native, invasive rust *M. hiratsukanum* as the species responsible for the epidemic outbreak that is raging on *A. incana* stands in the eastern sector of the Alps.

Outbreaks of native pathogens are common disturbances in forests, where they can also act as an important ecological factor of renewal of forest ecosystems and of maintenance of biodiversity. However, natural ecosystems can be seriously disrupted when a foreign, invasive pathogen is introduced. This is what seems to be happening in the eastern sector of the Alps. Everything indicates that the invasive, but now established and widespread rust *M. hiratsukanum,* can threaten the survival of *A. incana* along riparian formations, directly or indirectly.

## Figures and Tables

**Figure 1 jof-07-00617-f001:**
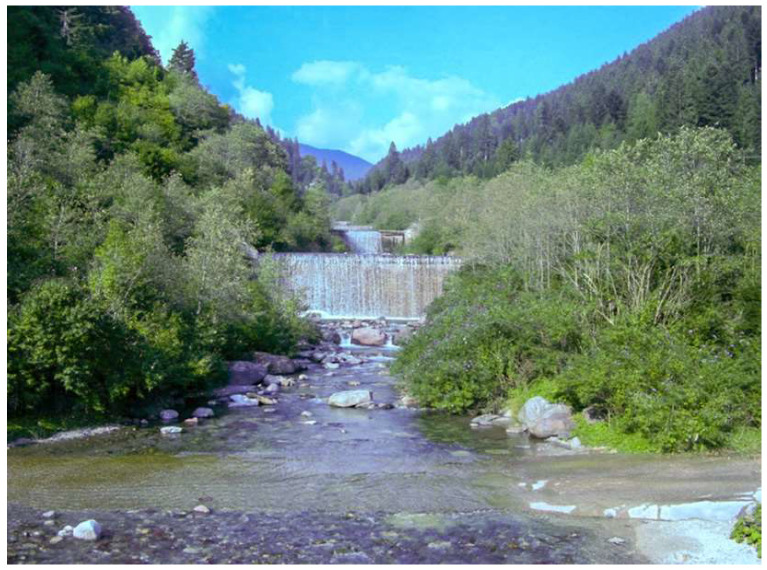
A mountain stream in Val Campelle (Trento province, Italian Alps). The riparian vegetation is dominated by *Alnus incana*, whose foliage, starting from the second half of July, appears heavily defoliated because of the attack by the rust pathogen *Melampsoridium hiratsukanum*.

**Figure 2 jof-07-00617-f002:**
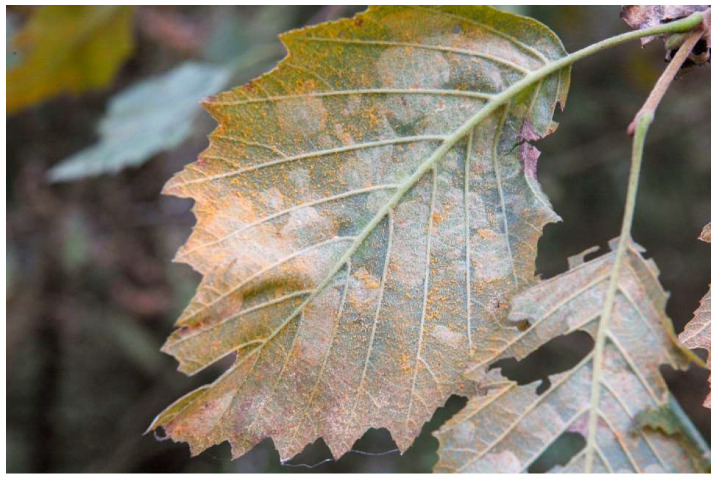
Powdery, orange-yellowish uredinia on the lower surface of *Alnus incana* foliage in July.

**Figure 3 jof-07-00617-f003:**
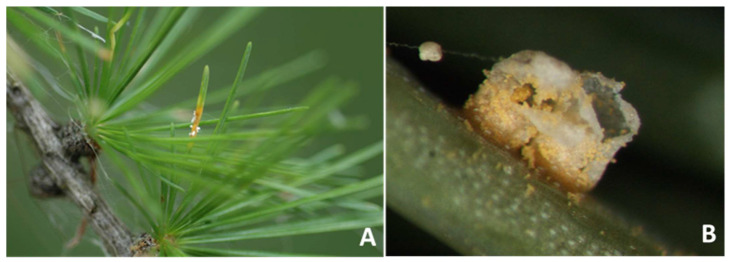
Foliage of *Larix decidua* with rust aecial sori. (**A**) A cluster of opened aecia, with the whitish peridium, clearly visible on a needle. The infected tissue area appears desiccated. (**B**) Particular of an erected, cylindrical, mature aecium. The peridium has already ruptured and aeciospores are being dispersed.

**Figure 4 jof-07-00617-f004:**
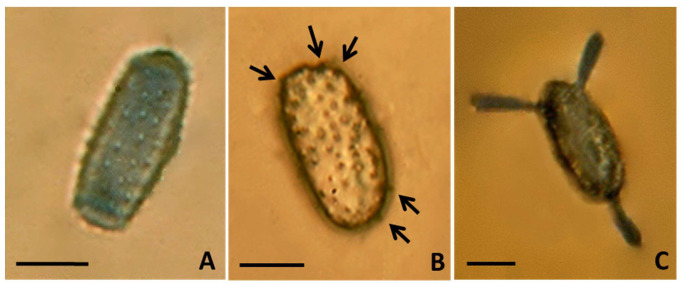
Light microscopy of *Melampsoridium hiratsukanum* uredinio-spores. (**A**) An oblong-obovoid uredinio-spore, with echinulae finely distributed all over the spore surface. Scale bar = 10 µm. (**B**) A mature uredinio-spore with a coarse, though uniform echinulation in evidence and up to five germ pores (arrows). Scale bar = 10 µm. (**C**) An uredinio-spore with three differentiating germ tubes. Scale bar = 10 µm.

**Figure 5 jof-07-00617-f005:**
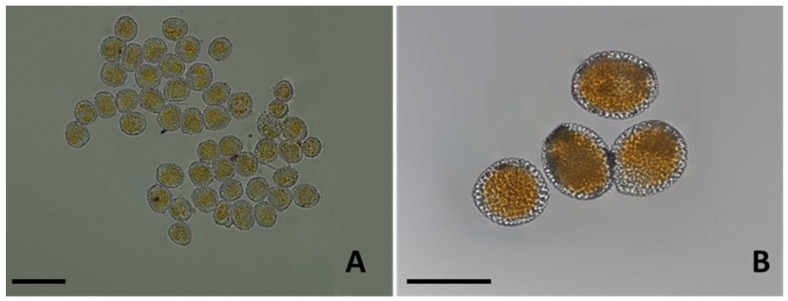
Light microscopy of *Melampsoridium hiratsukanum* aeciospores. (**A**) A group of aeciospores mounted in water, yellow-orange in colour. Scale bar = 40 µm. (**B**) Detail of four freshly harvested aeciospores with a marked orange colour and a visibly verrucose surface. Scale bar = 20 µm.

**Figure 6 jof-07-00617-f006:**
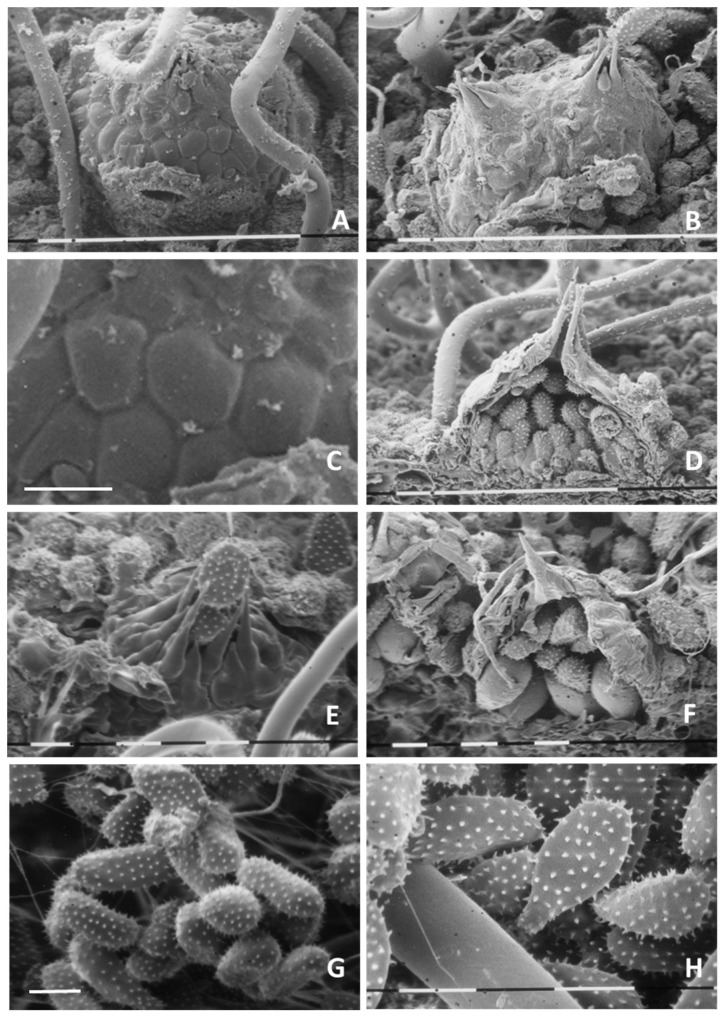
Ultrastructural imaging of the uredinial stage of *Melampsoridium hiratsukanum*. (**A**) A hemispherical uredinium, erupted through the leaf epidermis; a portion of the displaced epidermis bearing a stoma is noticeable. Scale bar = 100 µm. (**B**) A couple of gregarious sori with ostiolar cells extending into an acute apex. Scale bar = 150 µm. (**C**) Detail of Figure A: particular of the peridium composed of irregularly polygonal cells. Scale bar = 10 µm. (**D**) Cross section of a uredinium, with the very long, sharp spines, the thick, inner peridial wall and the internal uredinio-spores in evidence. Scale bar = 100 µm. (**E**) Uredinio-spores being expelled from an uredinium through a central pore made by ostiolar cells terminating in long, conical spines. Scale bar = 15 µm. (**F**) Cross section of an uredinium with clearly evident smooth, still immature uredinio-spores (arrow) in the lower portion of the sorus. Scale bar = 10 µm. (**G**) a group of finely echinulate uredinio-spores, with a rounded to obovoid form. Scale bar = 15 µm. (**H**) a group of finely echinulate uredinio-spores, from tear-shaped to clavate. Scale bar = 15 µm.

**Figure 7 jof-07-00617-f007:**
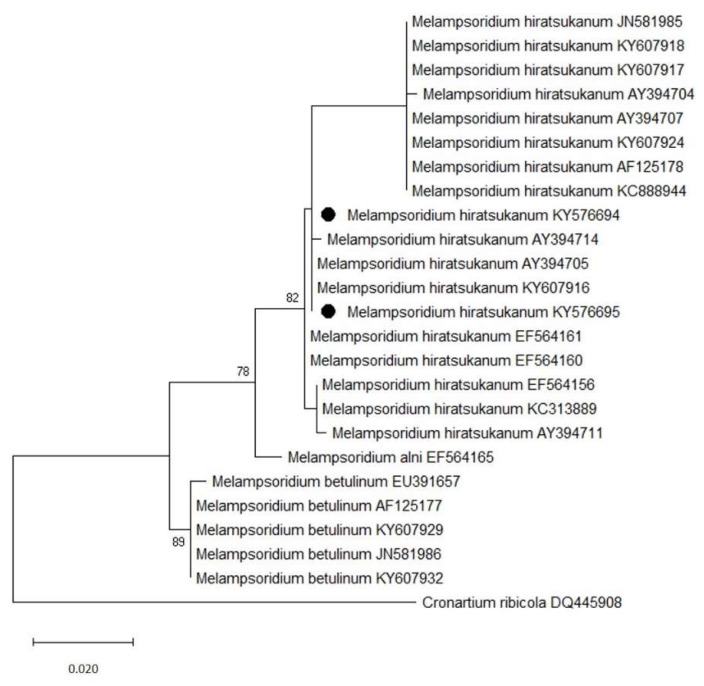
Phylogenetic tree based on Maximum Likelihood method of 24 rDNA ITS sequences of species of *Melampsoridium* from GenBank collection, identified by accession numbers, and rooted with *Cronartium ribicola*. Sequences from this study are indicated (●). Bootstrap values greater than 50% are shown above branches. Branch lengths are ML estimates and are scaled in terms of number of nucleotide substitutions per site. The analyses were conducted in MEGA X.

**Figure 8 jof-07-00617-f008:**
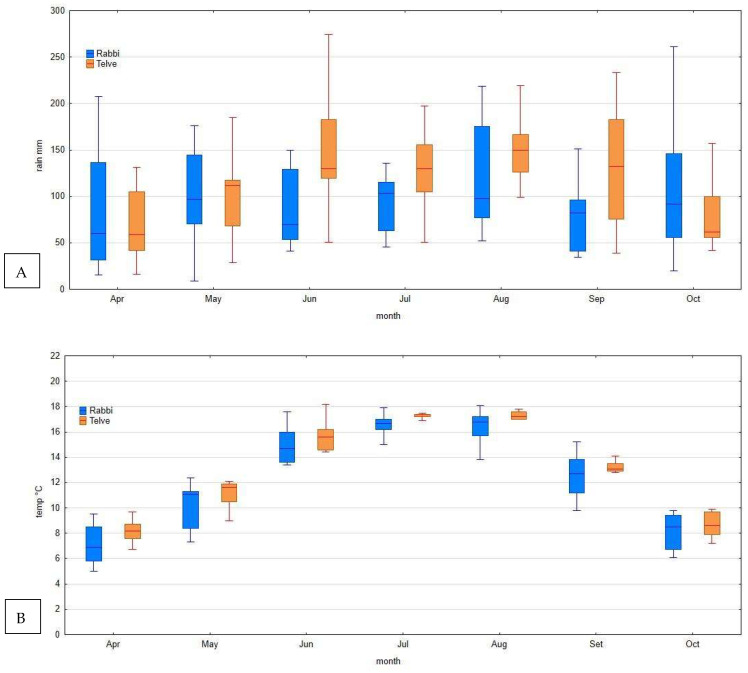
(**A**,**B**) Seasonal variation in precipitation (monthly total) (**A**) and air temperature (monthly average data) (**B**) measured at the Val di Rabbi and Telve meteorological stations (2010–2020).

**Figure 9 jof-07-00617-f009:**
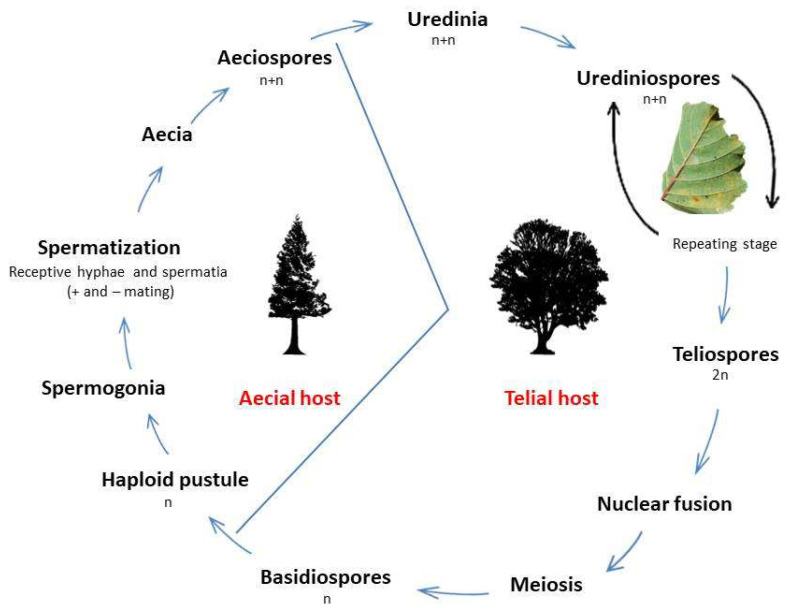
Life cycle of the hetero-macrocyclic rust *Melampsoridium hiratsukanum*, with details of its spore states and nuclear cycle on the aecial host (*Larix decidua*) and the telial host (*Alnus incana*).

**Figure 10 jof-07-00617-f010:**
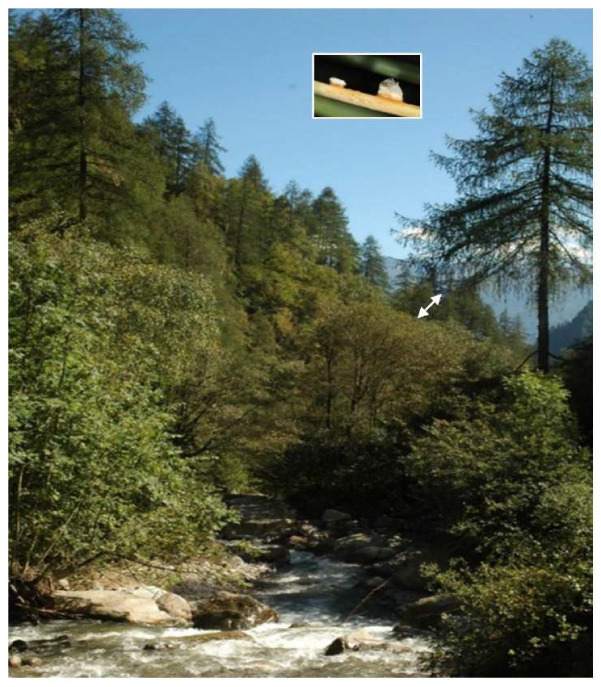
Typical alpine vegetation along a slope, with *Larix decidua* trees in a dominant position and the underlying riparian *Alnus incana*-dominated vegetation along a stream. The close proximity between the aecial (larch) and the telial (alder) hosts facilitated inoculum transmission (arrow); in the box: detail of a desiccated larch needle bearing two aecial sori: the whitish peridial wall of aecia (the largest is almost empty) clearly indicates that aeciospore have already been dispersed.

**Table 1 jof-07-00617-t001:** Key micromorphological characters for distinguishing the three species of *Melampsoridium* (*M. alni*, *M. betulinum* and *M. hiratsukanum*) reported on *Alnus* spp. and *Betula* spp.

Diagnostic Features	*Melampsoridium alni*	*Melampsoridium betulinum*	*Melampsoridium hiratsukanum*
Urediniospore morphology	From subfusoid to subclavate	From subfusoid to subclavate	Ovoid to oblong-obovoid
Urediniospore size (average, μm)	33.5 × 11.5	30.9 × 13.4	26.0 × 13.3
Distribution of echinulae	Absent at the two spore ends	Absent at the two spore ends	All over the spore surface
Length of uredinial ostiolar cells (range, μm)	20–25	20–25	30–45
Number and arrangement of germ pores	Two, one at each spore end	Up to six, on the two spore ends	Up to six, on the two spore ends

Morphometric data of *M. alni* and *M. betulinum* were from Hantula et al. [4], Roll-Hansen and Roll-Hansen [15], Kaneko et al. [19] and Kurkela et al. [20].

## Data Availability

All data generated or analysed during this study are included in this published article. The sequence data were deposited in the NCBI GenBank database.

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
