# Peer review of "First Documentation of Life Cycle Completion of the Alien Rust Pathogen Melampsoridium hiratsukanum in the Eastern Alps Proves Its Successful Establishment in This Mountain Range"

_jof, 2021, doi:10.3390/jof7080617_

Round 1

Reviewer 1 Report

This manuscript is a thorough examination of Melampsoridium hiratsukanum, using both morphological and molecular methods, to conclude that the pathogen is present in Italy and to link the aecial and uredinial hosts.  The study is comprehensive and my recommendation is acceptance following minor revisions.  The minor primarily grammatical revisions are noted below.

Title:  Italicize Melampsoridium hiratsukanum

Line 16:  The sentence is awkward suggest “Melampsoridium hiratsukanum is an alien rust fungus which has spread pervasively throughout several European countries following its introduction into North Europe at the end of the 20th century.”  The fact that it is heteroecious is noted in the first line of the Introduction so it can be removed from this sentence.

Line 259:  Replace “Infection interested” with “Infection affected” or “Infection occurred throughout the crown”.

Line 287:  Use “mid-“ rather than “half-“ to indicate the midpoint of the month.

Line 299:  Suggest “Morphological identification” as heading rather than “Conventional”.

Line 309: “Q1Urediniospore”? suggest “high” rather than “fairly good”

Line 324:  what is the range of aeciospore size (as per urediniospores given on line 306).

Figure 5:  Scale bars should be replaced with scale bars as per Figure 4.

Line 382:  Suggest “sequences of Melampsoridium spp.”

Line 407:  Suggest replace “resulted” with “were”.

Line 430.  Remove “anyhow”; suggest “LM examination of germinating urediniospores revealed the studied rust had multiple germ pores unlike M. alni that has a single germ pore at the end of each urediniospore.”

Line 452.  Suggest replace “this is” with “the peridium”

Line 456.  Use “BLAST analysis” rather than “blasting”.

Line 458.  Suggest replace “finds a further” with “is further corroborated by”.

Line 484.  Suggest “Infection of larch needles was very sparse, making it difficult to observe during surveys, especially when it occurs on tall trees with a large crown.”

Lines 505 – 510.  Split the single sentence in to two sentences.  “…this invasive fungus.  This is different from... where the disease has a variable and erratic impact.”

Line 511:  remove “non-essentiality” instead “Larch infection is not required for survival”.

Line 518:  Suggest “and results in partial defoliation”

Line 519:  Suggest “trees generally show normal” (remove “a”).

Line 533:  Replace “favored” with “facilitated”

Line 535:  Use “Phytophthora spp.”

Line 555:  Use “ as the species responsible”, or “as the pathogen responsible”

Line 575:  Nice dedication.

Author Response

I do not reply point by point to your comments because I accepted ALL of them. Thank you very much for your comments and suggestions that have undoubtedly helped improve this manuscript!

Reviewer 2 Report

In this study the authors characterize the life cycle of M. hiratsukanum, describe it morphologically and include some molecular data. This work can be of great interest to distinguish morphologically the Melampsoridium species. However, it needs some improvements before it can be published.

In general, molecular studies are very poor. Despite having sampled for several consecutive years, the authors only sequence two specimens. If, as the authors say in the introduction, several species of Melampsoridium have been confused and could infect the same hosts, it would be advisable to analyze a larger number of specimens to confirm that the infections observed are due to a single species. Furthermore, it would be advisable to amplify some other loci, since in the presented phylogeny, based on ITS rDNA, the different species do not form well supported clades. If the taxonomy of the group is doubtful, an exhaustive molecular study is necessary to confirm that M. hiratsukanum, M. betulinum and M. carpini are really different species.

Both the methodology and the results of molecular studies should be more detailed. In Material and Methods, the selection of taxa included in the phylogeny should be detailed. The program used for aligning the sequences and the program used to estimate the ML analysis with the parameter set should be specified. In Results, the number of positions included in the alignment, informative positions, likelihood value of the tree, should be added.

In the conclusions the authors affirm that there are enough characters to distinguish the species. I suggest they include a table comparing the three species.

The figures should be improved.

Fig. 1. Please, try to correct the color of the image. It has a blue dominant and the defoliated tree are difficult to observe.

Fig. 2 has poor quality, the uredinia are not in focus.

Fig. 4 has low resolution

Fig. 5. The scale is difficult to observe. Please, add a thicker line

Fig. 6. Use the same thickness for the scales in all photos

Other comments:

Please, write the complete name of the genera after dot.

The range of variation of Uredinia is very large (50-100 um). I suggest to indicate the measures as (minimum–){X −SD}−{X +SD}(–maximum), where X is the arithmetic mean and SD the corresponding standard deviation.

Author Response

(in bold our responses)

Comments and Suggestions for Authors

In this study the authors characterize the life cycle of M. hiratsukanum, describe it morphologically and include some molecular data. This work can be of great interest to distinguish morphologically the Melampsoridium species. However, it needs some improvements before it can be published.

Thank you for your appreciation of our manuscript. We have accepted almost all of your suggestions and modified our manuscript accordingly. We have also tried to explain our position on the molecular part. We thank you for your invaluable contribution which certainly made this manuscript to appear in a better light!

In general, molecular studies are very poor. Despite having sampled for several consecutive years, the authors only sequence two specimens. If, as the authors say in the introduction, several species of Melampsoridium have been confused and could infect the same hosts, it would be advisable to analyze a larger number of specimens to confirm that the infections observed are due to a single species.

The molecular analysis was restricted to a few specimens, mainly due to the difficulty to sample and extract fungal DNA from the inconspicuous, very sparse, small aecia found on the needles of Larix decidua. Instead, a large number of specimens were observed in the morphological studies. The better obtained sequence were published.

Many of the older reports that species of Melampsoridium can invariably infect the same host are wrong. These reports are a consequence of species misidentification, that has generated much confusion (as we have clearly written in the Introduction). Our investigations on M. hiratsukanum suggest the host range of this species to be narrower. In fact, as we have reported in the manuscript (lines 470-478, this revised version) neither Betula pendula, nor Alnus glutinosa, nor Alnus viridis trees growing in close proximity to infected Alnus incana trees, had any signs of infection by Melampsoridium hiratsukanum.

Furthermore, it would be advisable to amplify some other loci, since in the presented phylogeny, based on ITS rDNA, the different species do not form well supported clades. If the taxonomy of the group is doubtful, an exhaustive molecular study is necessary to confirm that M. hiratsukanum, M. betulinum and M. carpini are really different species.

We agree that the phylogenetic analysis is poor and, for this reason, we have moved the phylogenetic tree to "Supplementary Material". It was a former referee who insisted and demanded that we do a phylogenetic analysis! Our manuscript in centred on Life cycle, Morphology and Ecology of M. hiratsukanum. The molecular analysis of fungal DNA was not the main investigation of this study, as specified in the objectives. We now decided to maintain it here in the "Supplementary Material" solely as a complementary approach to confirm the morphological identification and the taxonomic affiliation of this rust into the species M. hiratsukanum. We have also clearly written in the text that the distinction between M. hiratsukanum and the other two species reported to infect Alnus spp. and Betula spp. does not rely on the phylogenetic analysis (lines 483-485, this revised version).  

Both the methodology and the results of molecular studies should be more detailed. In Material and Methods, the selection of taxa included in the phylogeny should be detailed. The program used for aligning the sequences and the program used to estimate the ML analysis with the parameter set should be specified. In Results, the number of positions included in the alignment, informative positions, likelihood value of the tree, should be added.

We have added some lacking information and, as we said above, put the phylogenetic tree as "Supplemetary Material", to give less emphasis to this part, which is nevertheless useful in confirming the identity of the sequences as belonging to M. hiratsukanum.

In the conclusions the authors affirm that there are enough characters to distinguish the species. I suggest they include a table comparing the three species.

Done. We have included a table where the main distinctive characters are summarized

The figures should be improved.

Done. We have used a graphics program to improve some tables (see below details)

Fig. 1. Please, try to correct the color of the image. It has a blue dominant and the defoliated tree are difficult to observe.

Done.

Fig. 2 has poor quality, the uredinia are not in focus.

Done.

Fig. 4 has low resolution

Unfortunately we have no better photos than these. We still made some improvements to the photos.

Fig. 5. The scale is difficult to observe. Please, add a thicker line

Done. We added a thicker line.

Fig. 6. Use the same thickness for the scales in all photos

Done. We have replaced the previous figure 6G (which had a very thick bar) with another which has a bar with a lesser thickness.

Other comments:

Please, write the complete name of the genera after dot.

Sorry, we can't understand what you mean here. To our knowledge, once a species has been mentioned a first time with its extended genus name, it can be quoted in the rest of the text (at least within the same Section) with the dotted genus name, followed by the species name.

The range of variation of Uredinia is very large (50-100 um). I suggest to indicate the measures as (minimum–){X −SD}−{X +SD}(–maximum), where X is the arithmetic mean and SD the corresponding standard deviation.

Ok, we re-checked the dimensions of Uredinia and have rephrased the sentence by reporting, mean, standard deviation and range.

Reviewer 3 Report

This is a well-written paper that presents interesting information about the ecological relevance of Melampsoridium hiratsukanum in Europe and presents evidence that the rust samples examined are in fact the invasive rust M. hiratsukanum.

General comments:

Some of the Genbank sequences that are most similar to the sequences acquired in this study are referred to as “voucher” sequences, but they appear not to include type material. The authors might want to address this issue of why type material has not been included in previous studies or the current study. Presumably, that material is not available, but it would still be valuable for the reader to know how certain the authors can be that these sequences and samples represent authentic M. hiratsukanum.

It is not clear why some type is in red. Perhaps this is a holdover from previous revisions.

Much of this story regarding morphology, ITS sequences, and the two alternate hosts was presented in Hantula et al. (Mycologia 2009). Although the Hantula et al. article is cited, the current authors should make clear what is new in the current study. In this context, while the discussion of the Hantula et al. paper in lines 479-481 is technically correct in saying M. hiratsukanum was not reported by Hantula et al. to be on Larix decidua in natural settings, this passage is misleading. My reading of Hantula et al. is that they did in fact convincingly link M. hiratsukanum samples from L. sibirica with nearby samples from Alnus incana.

Specific comments:

The bootstrap values on the phylogenetic tree are poor for the portion of the tree that includes M. hiratsukanum and M. alni. While the authors make a reasonably compelling argument that the samples studied here are M. hiratsukanum based on morphological and other data, they might want to acknowledge succinctly that the distinction between the two species does not rely on the phylogenetic analysis. It might also be worth pointing out specific ITS nucleotide positions that appear to distinguish between the two species.

Materials and Methods: Given the discussion about the need for proximity between the alder and larix hosts, it would be good for the reader to know where the urediniospores and aecia that gave rise to the Genbank sequences from this study were collected relative to each other.

Two paragraphs beginning line 250. Perhaps reiterate that this section is referring only to A. incana.

Lines 259-260: The following sentence is not clear. “Infection interested most of the crown, but its lower parts were the most severely affected; the upper parts of the crown were generally unaffected.” I suggest rewording it. Perhaps “interested” should be “involved”.

Line 287. Change “half” to “mid”. Also, is “courts” the correct word here?

Line 304: Perhaps change “as single” to “singly”.

Line 309: “Q1” delete?

Line 334: meaning of “firm”?

Line 354: meaning of “dwelled”?

Figure 7 legend. Italicize Melampsoridium?

Line 366: I suggest changing “Sequences search was” to “Sequence searches were”.

Line 408: Change “resulted” to “were”?

Line 430: I suggest changing “Anyhow” to the more formal “In any case”.

Figure 8 legend. Are temperature values medians? If so, perhaps add this to the legend.

Figure 9: Were the tree and leaf images used here created by the authors? Alternatively, are they known to be in the public domain?

Line 485: Change “interested” to “involved”.

Line 542: Add a space after the period in A.incana

Line 518: “substantiates” seems to be the wrong word here.

Line 527: perhaps change “on riparian” to “in the riparian”

Author Response

in bold our responses)

Comments and Suggestions for Authors

This is a well-written paper that presents interesting information about the ecological relevance of Melampsoridium hiratsukanum in Europe and presents evidence that the rust samples examined are in fact the invasive rust M. hiratsukanum.

Thank you for appreciation! As you can see below, we have accepted almost all your suggestions. We have also responded to some of your requests for clarification, hoping to have been clear enough and exhaustive. Thank you for helping us make this manuscript look better!

General comments:

Some of the Genbank sequences that are most similar to the sequences acquired in this study are referred to as “voucher” sequences, but they appear not to include type material. The authors might want to address this issue of why type material has not been included in previous studies or the current study. Presumably, that material is not available, but it would still be valuable for the reader to know how certain the authors can be that these sequences and samples represent authentic M. hiratsukanum.

We share your hypothesis on why type material has not been included in previous studies. We had considered those sequences because they were published in a peer reviewed journal (McKenzie et al. (2013), Plant Pathology & Quarantine - Doi 10.5943 / ppq / 3/2/1. However, they have now been deleted from our phylogenetic analysis.

It is not clear why some type is in red. Perhaps this is a holdover from previous revisions.

Yes, it is.

Much of this story regarding morphology, ITS sequences, and the two alternate hosts was presented in Hantula et al. (Mycologia 2009). Although the Hantula et al. article is cited, the current authors should make clear what is new in the current study. In this context, while the discussion of the Hantula et al. paper in lines 479-481 is technically correct in saying M. hiratsukanum was not reported by Hantula et al. to be on Larix decidua in natural settings, this passage is misleading. My reading of Hantula et al. is that they did in fact convincingly link M. hiratsukanum samples from L. sibirica with nearby samples from Alnus incana.

The novelty of this study is that we report for the first time the exotic rust Melampsoridium hiratsukanum on the native Larix decidua in Europe. In fact, Hantula et al. had found this rust to infect NATURALLY only the Asian larches L. dahurica and L. sibirica in the Tuusula area (Finland) or they managed to infect European larch (Larix decidua) following ARTIFICIAL INOCULATION with a rust fungus presumed to be M. hiratsukanum. However, no one before us has found European larch trees infected by this rust in NATURE. I have rephrased this issue in the manuscript to make it clearer.

Specific comments:

The bootstrap values on the phylogenetic tree are poor for the portion of the tree that includes M. hiratsukanum and M. alni. While the authors make a reasonably compelling argument that the samples studied here are M. hiratsukanum based on morphological and other data, they might want to acknowledge succinctly that the distinction between the two species does not rely on the phylogenetic analysis. It might also be worth pointing out specific ITS nucleotide positions that appear to distinguish between the two species.

Thank you for your suggestion. We have removed from the manuscript the part on molecular phylogeny, putting it only as Supplementary Material (Figure S1). We have also clearly written what you have suggested, namely that the distinction between the two species does not rely on the phylogenetic analysis. For your information: this part of phylogeny is a stretch that was asked by another reviewer.

Materials and Methods: Given the discussion about the need for proximity between the alder and larix hosts, it would be good for the reader to know where the urediniospores and aecia that gave rise to the Genbank sequences from this study were collected relative to each other.

Done

Two paragraphs beginning line 250. Perhaps reiterate that this section is referring only to A. incana.

Done

Lines 259-260: The following sentence is not clear. “Infection interested most of the crown, but its lower parts were the most severely affected; the upper parts of the crown were generally unaffected.” I suggest rewording it. Perhaps “interested” should be “involved”.

Done

Line 287. Change “half” to “mid”. Also, is “courts” the correct word here?

Done

Line 304: Perhaps change “as single” to “singly”.

Done

Line 309: “Q1” delete?

Done

Line 334: meaning of “firm”?

It was a typo. The term has been deleted

Line 354: meaning of “dwelled”?

The term has been corrected

Figure 7 legend. Italicize Melampsoridium?

Done

Line 366: I suggest changing “Sequences search was” to “Sequence searches were”.

Done

Line 408: Change “resulted” to “were”?

Done

Line 430: I suggest changing “Anyhow” to the more formal “In any case”.

The sentence has been reformulated by eliminating "Anyhow"

Figure 8 legend. Are temperature values medians? If so, perhaps add this to the legend.

Done

Figure 9: Were the tree and leaf images used here created by the authors? Alternatively, are they known to be in the public domain?

The images of the trees have been replaced with other images that are free on the web (site: www.123freevectors.com); the image of the alder leaf is ours

Line 485: Change “interested” to “involved”.

The sentence has been rephrased and "interested" has been deleted

Line 542: Add a space after the period in A.incana

Done

Line 518: “substantiates” seems to be the wrong word here.

Corrected

Line 527: perhaps change “on riparian” to “in the riparian”

Done

Round 2

Reviewer 2 Report

Nowadays, the molecular identification of fungi is essential to achieve a more realistic vision of their biodiversity. Therefore, the phylogenetic analyzes should remain within the manuscript, not as supplementary material, despite the molecular study not being the main purpose of the work

In “Material and Methods” section, "molecular identification", some details are still missing:

Which program was used to assemble and edit the sequences and to perform the alignments.

ITS rDNA alignments generally have ambiguous regions that must be selected and removed before conducting phylogenetic analyzes. The authors, however, do not mention this point.

I notice an incongruity between Material and Methods and Results. According to Material and Methods, the authors used a distance method to elaborate their phylogeny, line 234-235:

 “molecular phylogeny obtained by applying Neighbor-Join and BioNJ algorithms”.

But in Results they say “A phylogenetic tree based on maximum likelihood”.

If the authors have used a distance method to establish phylogenetic relationships, I would recommend them to apply a more robust method, based on evolutionary models (ML or Bayesian inference).

Bootstrap values are indicated in the phylogenetic tree, but this analysis is not mentioned in the methodology. Authors should specify the number of replicas on which this analysis is based.

Author Response

We have added what you requested. WITHOUT DOUBT, your requests for further "refinements" have contributed significantly to making this manuscript appear in its best light.  Sincere thanks!

(in bold our responses)

Comments and Suggestions for Authors

Nowadays, the molecular identification of fungi is essential to achieve a more realistic vision of their biodiversity. Therefore, the phylogenetic analyzes should remain within the manuscript, not as supplementary material, despite the molecular study not being the main purpose of the work

Ok, we brought the phylogenetic tree into the manuscript again.

In “Material and Methods” section, "molecular identification", some details are still missing:

Which program was used to assemble and edit the sequences and to perform the alignments.

These details have been added in Material and Methods. Sequence data were aligned and edited using clustalW implemented in MEGA X (Kumar S., Stecher G., Li M., Knyaz C., and Tamura K. (2018). MEGA X: Molecular Evolutionary Genetics Analysis across computing platforms. Molecular Biology and Evolution 35:1547-1549.).

ITS rDNA alignments generally have ambiguous regions that must be selected and removed before conducting phylogenetic analyzes. The authors, however, do not mention this point.

This detail has been added.

I notice an incongruity between Material and Methods and Results. According to Material and Methods, the authors used a distance method to elaborate their phylogeny, line 234-235:

 “molecular phylogeny obtained by applying Neighbor-Join and BioNJ algorithms”.

But in Results they say “A phylogenetic tree based on maximum likelihood”.

If the authors have used a distance method to establish phylogenetic relationships, I would recommend them to apply a more robust method, based on evolutionary models (ML or Bayesian inference).

It has been corrected. The analysis was done using Maximum Likelihood method. Initial trees were obtained for the heuristic search automatically by applying Neighbor-Join and BioNJ algorithms to a matrix of pairwise distances estimated using the Maximum Composite Likelihood (MCL) approach.

Bootstrap values are indicated in the phylogenetic tree, but this analysis is not mentioned in the methodology. Authors should specify the number of replicas on which this analysis is based.

It has been added in the Materials and Methods.